# Frequency Diversity Array for Near-Field Focusing

**Xu Han [1], Shuai Ding [1,*], Yongmao Huang [2,*], Yuliang Zhou [1], Huan Tang [1] and Bingzhong Wang [1]**

[1] Institute of applied physics, University of Electronic Science and Technology of China, Chengdu 611731, China; uestc_xuhan@foxmail.com (X.H.); zhouyl8988@hotmail.com (Y.Z.); uestctang@sina.com (H.T.); bzwang@uestc.edu.cn (B.W.)

[2] School of Electrical and Electronic Information, Xihua University, Chengdu 610039, China

\* Correspondence: uestcding@uestc.edu.cn (S.D.); ymhuang128@gmail.com (Y.H.)

**Abstract:** In this study, a numerical optimization method is proposed to achieve the near-field focusing of square arrays and circular arrays. This method introduced the frequency diversity array (FDA) approach to change the initial amplitude and working frequency. By adjusting the working state of each antenna, the field distribution on any plane can be artificially controlled. To analyze the FDA, a mathematical model for the FDA has been built and the model has been optimized by a numerical algorithm. The results of two different kinds of arrays are verified by numerical methods and full-wave simulation.

**Keywords:** frequency diverse array (FDA); time-varying; focusing; optimization

## 1. Introduction

Wireless power transmission (WPT) technology can be divided into far-field focusing technology and near-field focusing technology. The development of far-field focusing technology is already quite mature [1]. Near-field focusing refers to the concentration of waves and the convergence of energy in the near-field range and there is very little research in this area.

Near-field focusing is a hot issue. In the past, people always used arrays to achieve near-field focusing. A focusing array can focus the energy from an antenna to a small point. However, with the development of technology, near-field focusing has more application scenarios, such as radiofrequency identification readers, medical applications, noncontact microwave sensing and microwave wireless power transmissions; traditional methods are no longer enough to meet people's needs.

The frequency diversity array (FDA) has attracted increasing attention because of its unique time-varying, distance-varying and angle-varying spatial focusing characteristic. Although more people are investigating this issue, near-field focus is still a difficult problem. The FDA [2] introduces a small-frequency offset between the array elements, so that the focus spots are not only related to the angle, but also the distance and time, which are three-dimensionally related. Since it has one more degree of freedom, it has an advantage in the two-dimensional joint detection of targets [3–5].

In 2006, the FDA was first proposed by Antonik [6], marking the beginning of FDA studies. This new array introduced a small-frequency offset between adjacent antenna elements, unlike a traditional phased array [7]. The offset in frequency enables periodic spatial beam scanning [8]. Through reasonable algorithm selection and parameter design, and some improvements to the objective function, the FDA can obtain a smaller focusing spot [9] compared to the existing arrays. With the smaller focusing spot, we can apply this feature to wireless power transmissions and secure communications [10–12]. Therefore, this paper focuses on the optimization of antenna arrays, aiming to use the unique space-time correlation of FDA to solve the problem of expensive phase shifters in traditional phase arrays.

In this study, two mathematical models for two different FDAs are discussed. To evaluate the focusing effect, we used the genetic algorithm (GA) to optimize its parameters and then verified the results with numerical methods and the full-wave simulation. The numerical results match the simulation results very well. To emphasize the importance of optimization, the before and after optimization results of the half-power beam-width and sidelobe are compared. Our results show that this is an efficient near-field focusing method. Not only that, the focusing effect achieved in this paper is more excellent than other literatures, whether it is the side lobes of focusing spot or the half power width of the focusing angle. More importantly, the optimization method proposed in this article has a bandwidth of 10%, which most articles do not focus on [13,14].

## 2. Near-Field Focusing of Square-Distributed Arrays

### 2.1. Mathematical Modeling for Square-Distributed Arrays

First, we concentrate on the near-field focusing characteristics of a square-distributed array, which is shown in Figure 1 [15,16]. The array elements are regularly distributed on the plane, and the operating frequency and relative amplitude of the array, are controlled to achieve a better focusing effect in the near field. To obtain the parameters of the array antenna, the genetic algorithm is used with the formulas of the FDA.

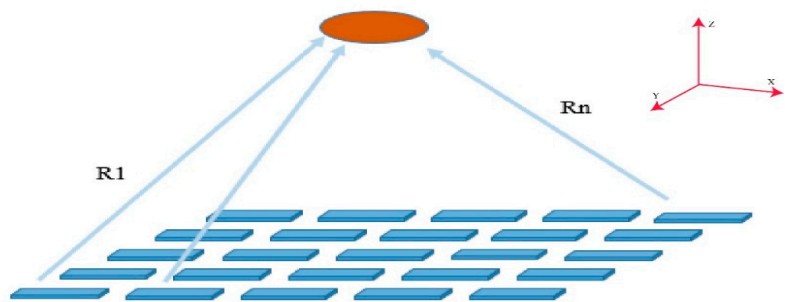

**Figure 1.** Rectangular distribution array diagram.

For a two-dimensional array, since the radiation fields of the array elements at the focus position need to superposed at the same phase, the function of the radiation field of the array must eliminate the phase-shift factor. In order to achieve phase control under the frequency varying conditions, the following formulas for the spatial field distribution of the arrays were adopted to optimize the energy focusing point [17]:

$$E = \sum_{m=1}^{N} \sum_{n=1}^{N} A_{mn} \alpha_{mn} e^{j2\pi((f_0 + \Delta f_{mn})(t - \frac{r}{c}) + \varphi_{mn})} \tag{1}$$

$$P = |E|^2 \tag{2}$$

$$\varphi_{mn} = 2\pi \frac{R_{mn}(f_0 + \Delta f_{mn})}{c} \tag{3}$$

$$R_{mn} = \sqrt{x_{mn}^2 + y_{mn}^2 + z_0^2} \tag{4}$$

$$\Delta f_n = \frac{g(n) - (n \sin \theta_0)/2}{t - R_0/c + n \sin \theta_0/(2f_0)} \tag{5}$$

$$E = \sum_{n=-(N-1)/2}^{(N-1)/2} e^{j(2\pi f_0(t - \frac{R}{c}) + 2\pi f_0(\frac{nd \sin \theta}{c}) + 2\pi n\Delta f(t - \frac{R}{c}) + 2\pi n\Delta f \frac{nd \sin \theta}{c} + \varphi_n)} \tag{6}$$

where $E$ represents the electric field, $f_0$ represents the center frequency, $C$ is the speed of light in a vacuum, $N$ is the number of array elements of the square array, $A_{mn}$ is the amplitude of the array

element, $\alpha_{nm}$ is the attenuation factor and $\Delta f_{mn}$ is the offset of the array element from the center frequency. $R_{mn}$ is the distance from the target point to the array element and $x_{mn}$, $y_{mn}$ and $Z_0$ represent the location of the array element. The most important parameter in the above formula is $R_{mn}$. Different initial phases are applied according to the different $R_{mn}$. The array elements, which applied different initial phases, will all focus on the same target point to achieve the effect of near-field focusing.

To obtain the focus at the specified position, the above parameters need to be optimized. The corresponding parameter values can be obtained by genetic algorithms. In the near-field focus optimization process of the array, to reduce the complexity, we remove the complex frequency control function and apply a constant operating center frequency to each array element. After performing the above operation, the parameters that need to be optimized are only $A_{mn}$ and $\Delta f_{mn}$.

With the above formula and operation, we still need the optimization objective function. Only the minimum value of the optimization function needs to be obtained in the optimization process, and the optimization parameters can be obtained by the GA. The optimization objective function is as follows:

$$\psi = \sum_{t=0}^{t=N} \left( W_1 \left( \frac{|X_{max} - X_0|}{X_0} + \frac{|Y_{max} - Y_0|}{Y_0} \right) + W_2 |HPBW(X_0, Y_0, t)| + W_3 |SLL(t)| \right) \tag{7}$$

In the above formula, $X_0$ and $Y_0$ represent the matrix row and column values at the target point in the optimization plane. $X_{max}$ and $Y_{max}$ represent the maximum power in the plane. Formula 5 ensures that the power value at the target point is the maximum power in the plane. HPBW($X_0$,$Y_0$,$t$) represents the half-power beam-width in two directions of the target focus point. SLL(t) represents the sidelobe of the plane. $W_1$, $W_2$ and $W_3$ are the parameters' weights.

## 2.2. Simulation Results

In the optimization process, a square array with 25 array elements is established [18–21]. The central operating frequency is 2.4 GHz. The 10% bandwidth of the optimization frequency is set, and the frequency range can be selected from 1 GHz to 3 GHz. The distance between adjacent array elements is set as 0.8 times of the wavelength, which is

$$d = 0.8 \times \frac{c}{f_0} = 0.1m \tag{8}$$

The target point is set at 1.5 m on the central axis of the array. Since the central wavelength of the array element is 0.125 m when it works, the 1.5 m position can be approximately considered as the far field with respect to wavelength, but if we take the far-field judgment formula into consideration:

$$Z = \frac{2D^2}{\lambda} \tag{9}$$

then the 1.5 m position will still be near field.

In the optimization plane, both the length and width of the optimized plane are 0.6 m. In this work, the GA of the intelligent optimization algorithm was used next. To get a better result as fast as possible, we set the number of the population size to 30, the maximum number of iterations to 30, the optimization time to 5 ns and the sampling interval to 0.1 ns. In this optimization process, we also set $W_1$, $W_2$ and $W_3$ to 1. As it is a square-distributed array, the array elements are symmetrical about the center of the array. To reduce the number of optimization parameters and maintain the symmetry, the same optimization parameters are used for the array elements, which has the characteristic of center symmetry in the optimization process. Therefore, although there are 25 array elements, only five parameters need to be optimized. After utilizing the above settings, we obtained the results shown in Table 1.

**Table 1.** Frequency offset and the amplitude offset.

| Index | $\Delta f_{mn}$ | $A_{mn}$ |
|---|---|---|
| 1 | 1.22 | 1.00 |
| 2 | 0.87 | 0.32 |
| 3 | 1.49 | 0.19 |
| 4 | 1.04 | 0.02 |
| 5 | 2.18 | 0.11 |
| 6 | 0 | 0.11 |

Taking the parameters above into account, we obtained the following focus effect figures [22–25]. The results were calculated by numerical methods.

Figure 2 shows that, although the sidelobe is very high, which is mainly because the three optimization goals in the previous article are set to equal weights, it has a good focusing performance after stabilization, which proves that the square distribution FDA realizes the near-field focusing [26,27].

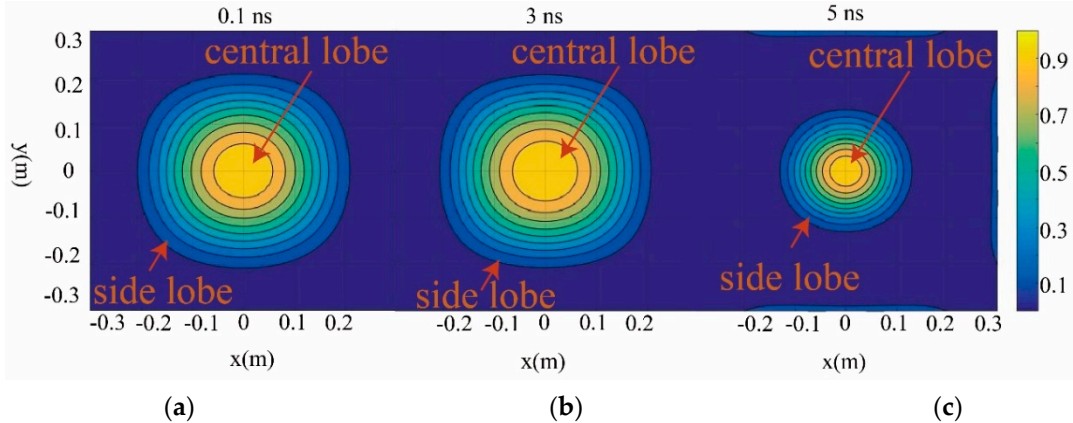

**Figure 2.** Array focus effect figures at different moments: (**a**) top view at t = 0.1 ns, (**b**) top view at t = 3 ns and (**c**) top view at t = 5 ns.

To highlight the importance of optimization, the half-power beam-width and side lobe size of the FDA array before and after optimization are compared.

Figure 3a shows the change in the half-power beam-width before and after optimization with time. Figure 3a shows that the half-power beam-width of the optimized array is not only smaller than that before optimization, but it is also more stable. The half-power beam-width before optimization is not only larger, but also has the characteristic of periodic change. Figure 3b shows the change of the side lobes of the array before and after optimization with time. Since the range of the side lobes is too large, the ordinate is represented by the logarithm. It can also be seen that the side lobes of the array before optimization are too large, unstable, and that they periodically change. The optimized array has smaller side lobes and is more stable.

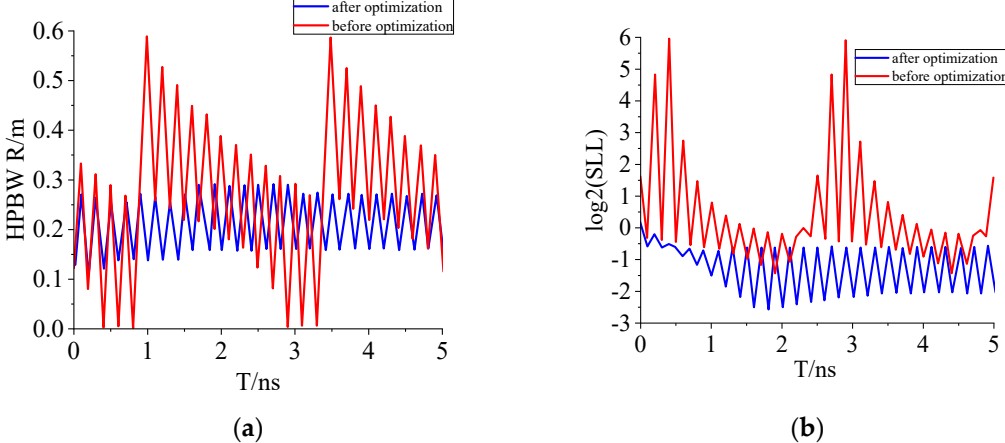

**(a)**                                                    **(b)**

**Figure 3.** (**a**) Half-power beam-width with time and (**b**) sidelobe changes with time.

The simulation results and the comparison with the pre-optimized array indicate that the rectangular distribution array optimized by the process is far superior to the unoptimized array, in both the focus stability and the focus spot parameters.

## 3. Near-Field Focusing of the Circular Array

### 3.1. Mathematical Modeling for Circular Distributed Arrays

We now discuss the optimization of a circular distribution array. To facilitate the comparison with the square array, the array was designed to make the area difference between the circular array and the square array as small as possible. The schematic diagram of the circular distribution array is shown below in Figure 4.

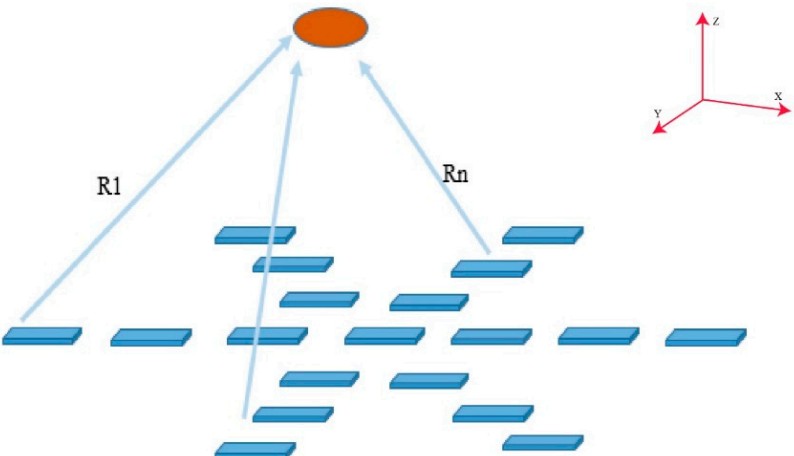

**Figure 4.** Circular distribution array diagram.

To calculate the field formula of the circular distributed array, the coordinates of array elements should be calculated first. For the convenience of calculation, the coordinates of the circular distributed array can be transformed into a rectangular coordinate system as follows:

$$x = R \times \sin(\theta)$$
$$y = R \times \cos(\theta)$$

(10)

The rest of the calculation is the same as that for the rectangular-distributed array.

### 3.2. Simulation Results

For the circular distributed array, 19 elements are distributed symmetrically in the center, as shown in Figure 4, and the angle between the adjacent array elements is 60°, which ensures that the distance between the adjacent elements is the same. The distance between the array elements is also 0.8 times that of the wavelength, the optimization objective function used is the same as Equation (5), and other simulation settings used are the same as that of the rectangular distributed array. The only difference is that for the circular distribution array arranged as shown in Figure 4, 6 $\Delta f_{mn}$ and 7 $A_{mn}$ need to be optimized. The optimization results are shown in Table 2.

**Table 2.** Frequency offset and the amplitude offset.

| Index. | $\Delta f_{mn}$ | $A_{mn}$ |
|--------|------|------|
| 1 | 0.90 | 0.45 |
| 2 | 1.60 | 0.18 |
| 3 | 1.68 | 0.27 |
| 4 | 2.15 | 0.26 |
| 5 | 1.33 | 0.92 |
| 6 | 1.07 | 0.94 |
| 7 | 0 | 0.02 |

After optimization, the final parameters above were applied, and we obtained the focus effect figures shown in Figure 5. The results were calculated by numerical methods.

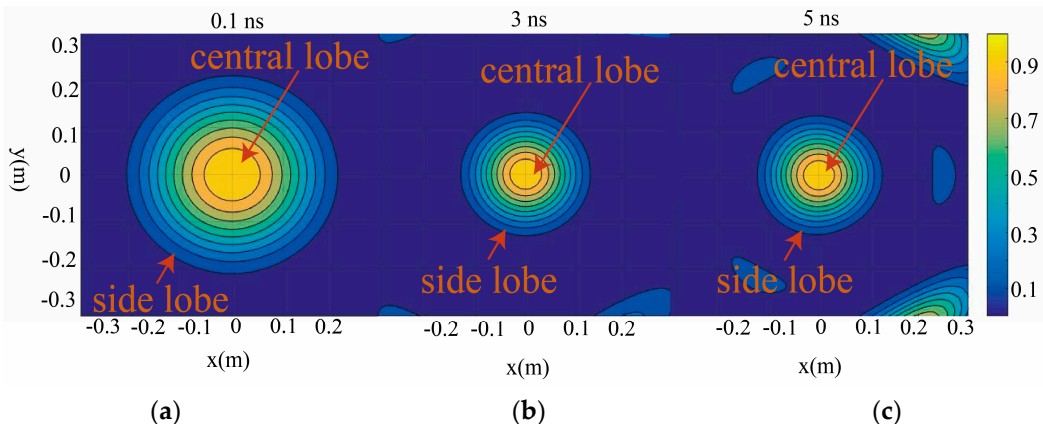

|     |     |     |
|-----|-----|-----|
| (a) | (b) | (c) |

**Figure 5.** Array focus effect figures at different moments: (**a**) top view at t = 0.1 ns, (**b**) t = 3 ns and (**c**) t = 5 ns.

Figure 5 shows that in the beginning period of optimized focusing, the focusing effect is relatively poor. With the passage of time, the focusing performance at the target point tends to be stable. These properties are the same as those of the square array, but compared with the square array, the focusing effect of the circular array is worse. Figure 5 shows that a larger energy distribution appears at the edge.

Next, we compare the half-power beam-width and side lobe size.

Figure 6a shows the change in the half-power beam-width of the circular and square arrays with the same sampling points. Figure 6a shows that the half-power beam-width of the square distribution array is slightly better than that of the circular distribution array. Figure 6b shows the change in the sidelobe with the same sampling time of the circular and square arrays. The sidelobe parameters of the square arrays are not different from those of the circular arrays at the initial stage, but the parameters of the square arrays are better than those of the circular arrays as time goes on. However, this does not mean that the circular distribution array is necessarily worse than the square distribution array. Although the areas of the two arrays are roughly the same, the number of elements in these

arrays is different. The rectangular array has 25 elements, while the circular distribution array only has 19 elements. A greater number of elements means there is a stronger spatial field distribution regulation performance. This is an advantage of the square-distributed array. However, if the number of elements is the same, the circular-distributed array must have a larger area, and even if the number of circular array elements is less than that of the square array, the number of optimization parameters in the optimization process is larger than that of the square array. In general, between the two optimized arrays discussed above, the near-field focusing performance of the square array is slightly better than that of the circular array.

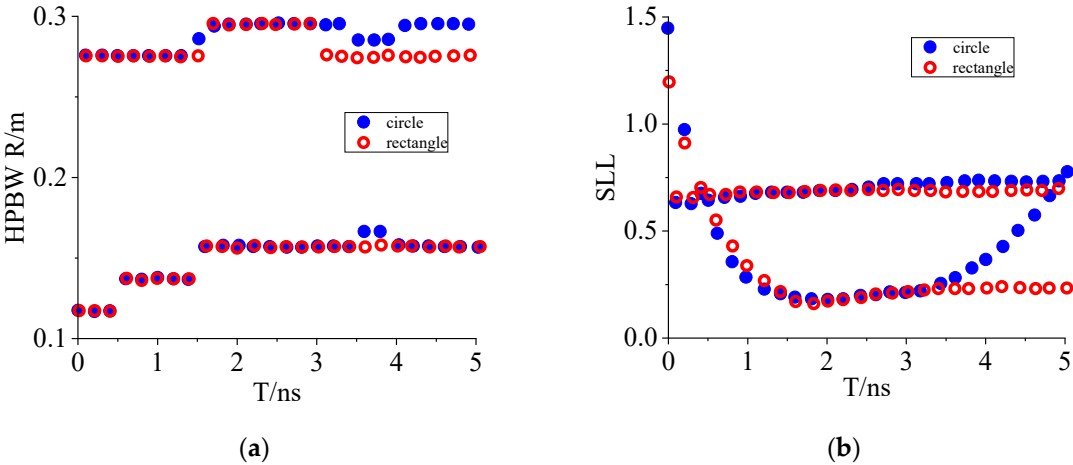

(**a**)

(**b**)

**Figure 6.** (**a**) Half-power beam-width with time and (**b**) sidelobe changes with time.

In fact, the above optimization method is not only used in square array and circular array, but is also applicable to the array optimization of any geometry. Therefore, in this paper, a universality and general optimization method has been proposed.

## 4. Simulation Verification of the Full Wave Simulation

After we performed the optimization and simulation process mentioned above, and the comparison analysis of the array, we verified the focusing characteristics of the pre-optimized and optimized FDA. This will be discussed in this section. The simulation method mainly discussed here is a full wave simulation. For the square array simulation verification, the first step is to construct a square array in the full-wave simulation; the simulation array [28] is shown in Figure 7.



**Figure 7.** Square array.

Twenty-five identical elements are set. Each element is set as a dipole with a length of 5.4 cm. The S-parameter of the elements is shown in Figure 8.

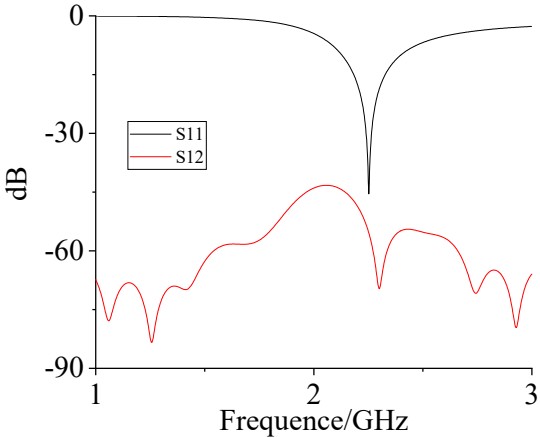

**Figure 8.** S-parameters of the square array's element.

The working frequency band of the square array is 2.4 GHz–2.62 GHz, and we can see from Figure 8 that the array elements' S11 are all below −10 dB.

This S-parameter is an active S-parameter and its calculation method itself has taken the coupling between the array elements of the antenna array into account. We can see from Figure 8 that, even considering the coupling between the array elements of the antenna array, the dipole's S11 is less than −10 dB within the operating bandwidth, and its coupling with other adjacent array elements, which is shown in the Figure 8 S21 curve, is less than −45 db, thus there is basically no coupling energy, indicating that this dipole is very suitable for verifying the algorithm in this paper. According to the 3 dB bandwidth method, which definition is:

$$Q = \frac{f_0}{B} \tag{11}$$

where $f_0$ is the center frequency and $B$ is the 3 dB bandwidth, we calculate that the Q factor [29–31] of the dipole is 2.3.

After the experimental model was designed, different signal [32] sources were applied to each array element according to the frequency and amplitude parameters obtained after the square array optimization, as shown in Figure 9.

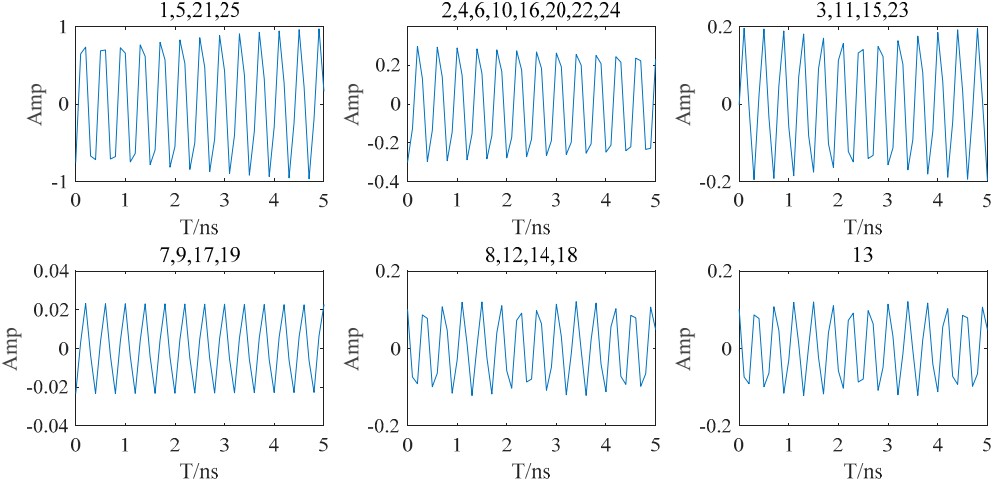

**Figure 9.** Excitation signals of the elements.

To monitor the field intensity distribution of the optimized plane, a plane monitor was set at 1.5 m above the array. The size of the monitor was 0.6 m × 0.6 m. After the above simulation settings, the next simulation verification stage was carried out. The comparison diagram of simulation results is shown in Figure 10.

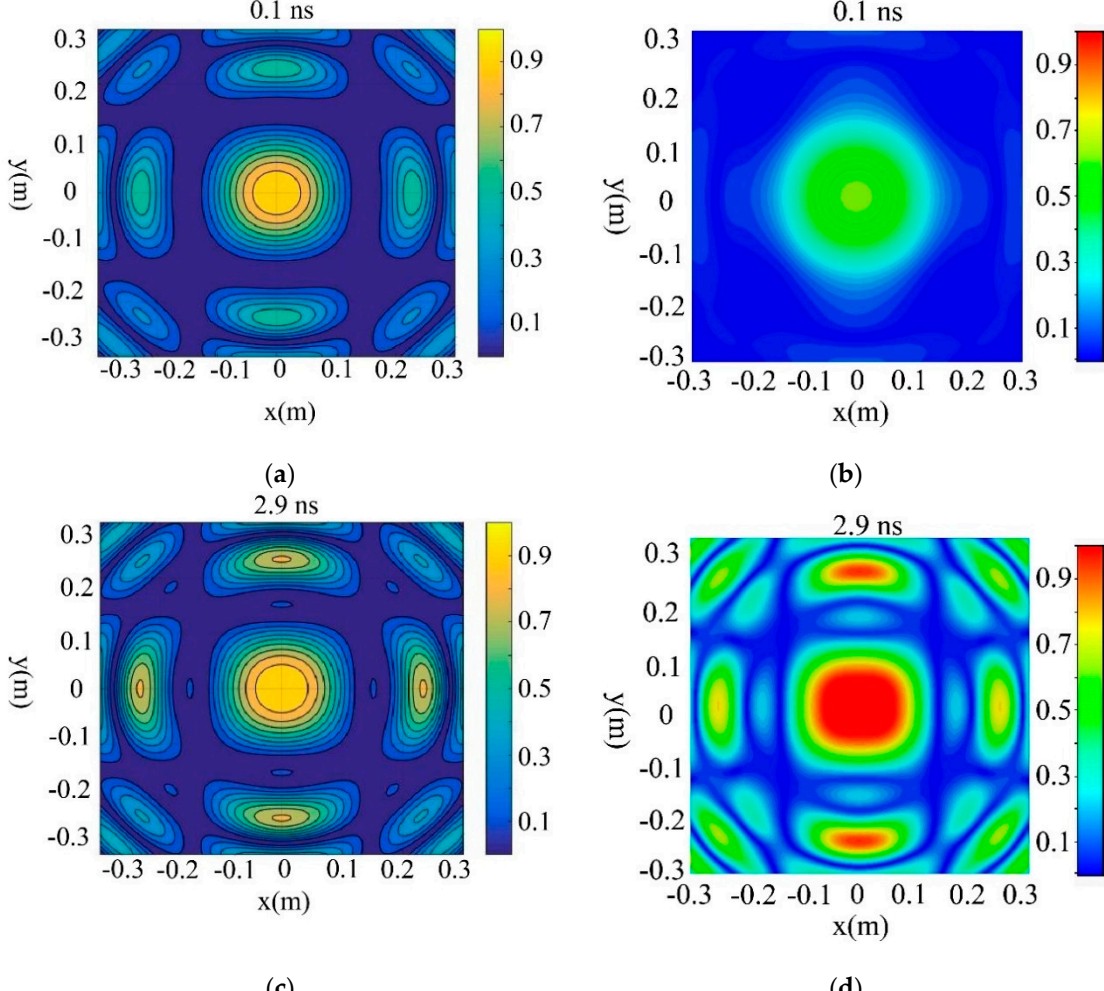

**Figure 10.** Array focus effect figures with different moments and different methods: (**a**) field distribution at 0.1 ns in the numerical methods after optimization, (**b**) field distribution at 0.1 ns in the full wave simulation after optimization, (**c**) field distribution at 2.9 ns in the numerical methods after optimization and (**d**) field distribution at 2.9 ns in the full-wave simulation after optimization.

In Figure 10, because the simulation diagram has been normalized, Figure 10a,c look very similar, but their corresponding amplitude values are quite different, which can also be seen from the corresponding full-wave simulation diagram. We can see, however, from the figures, that the focus spot changed from circular to oval. This is mainly because of the fact that, when formulating the near-field focus of the array, we consider 1.5 m as the far-field of the array and discarded the term of the field direction, resulting in that the optimization results did not completely match the experimental results. All this can be explained as the direction of the field is different from the angle of intersection of the field and the optimized plane. This figure shows that the full wave simulation results are basically consistent with the numerical methods results. The focusing performance of the optimized square distribution array is much better than that of the unoptimized array.

To show the importance of optimization, the comparison chart of the half-power beam-width over time, before and after optimization, is given below.

Figure 11 shows that the optimized array focusing parameters are significantly better than those before optimization and, in fact, some elements have too small amplitude, and after simulation verification, which is shown in Figure 12, removing them has no effect on the focusing effect.

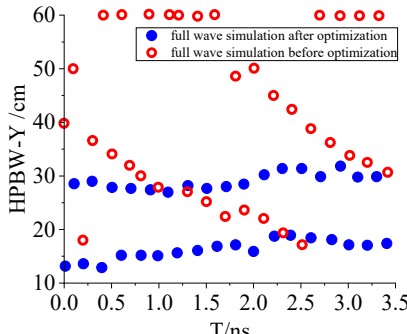

**Figure 11.** Comparison of the rectangular array's half-power beam-width before and after optimization.

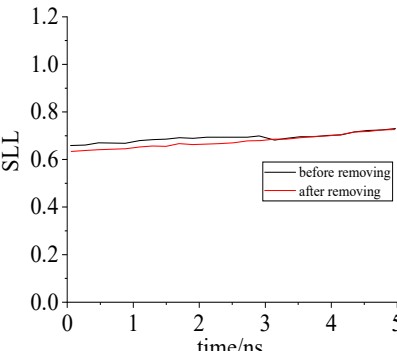

**Figure 12.** Comparison of before and after removing the unimportant elements.

## 5. Conclusions

In this study, we mainly discussed the near-field focusing characteristics of two distributed arrays, which gives a theoretical basis for applying the FDA in the field of wireless energy transmission [33,34]. In the first part, the spatial field distribution and power distribution formula of the square distributed array were given first. Next, the initial phase of the excitation signal applied to the array element was calculated according to the distance between the array elements, the focus of the target, and the working frequency. Then, the complete excitation of the array element was formed. Using the greedy optimization of the variables, the two parameters are optimized by the genetic algorithm to get the optimized parameters. Finally, we discussed the simulation of the near-field focusing performance of the optimized square array. We also verified the focusing performance of the optimized square distribution array. In the second part, we compared a circular array with the square array. When the area difference between the two kinds of arrays is small, the performance of the square array is slightly better than that of the circular array. Then we discussed the verification of the correctness of optimization using the full-wave simulation. The simulation results of the full wave simulation are basically the same as those of the numerical methods. The focus graph of the three moments in the focus time was extracted for comparison, and it proves the importance of array optimization.

**Author Contributions:** Conceptualization, X.H. and S.D.; investigation, X.H. and H.T.; writing—original draft preparation, X.H., S.D. and Y.H.; writing—review and editing, Y.Z. and B.W. All authors have read and agreed to the published version of the manuscript.

**Funding:** This work was supported by National Natural Science Foundation of China under Grant No. 61601087, Fundamental Research Funds for the Central Universities under Grant No. ZYGX2019Z016 and the Sichuan Science and Technology Program under Grant Nos. 2018GZ0518, 2020YJ0273 and 2020YFS0358.

**Conflicts of Interest:** The authors declare no conflict of interest.

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
