# Peer review of "Frequency Diversity Array for Near-Field Focusing"

_electronics, doi:10.3390/electronics9060958_

Round 1

Reviewer 1 Report

The paper presents a numerical optimization method for the near-field synthesis of square and circular antenna arrays.  The method exploits the frequency diversity array (FDA) approach to modify the amplitude of the excitations and the working frequency of the array. Numerical simulations are carried out to investigate the performance of the considered method.

The paper is readable and properly organized, but the introduction of the problem and the overall contribution are unclear. Furthermore, the application domain of the discussed algorithm seems extendable. Thus, the referee suggests a revisiting of the manuscript. The main issues are listed in the following of this review.

1) The Introduction should be modified to better identify the analyzed issue. More precisely, the authors should specify: (i) what is their contribution in terms of the algorithm, since currently the paper seems simply the application of an existing approach to the antenna research field; (ii) how this contribution locates with respect to the existing approaches for near-field synthesis.

2) A satisfactory literature overview on near-field synthesis is not present. many papers have analyzed this topic, such as: (i) “Pattern synthesis of array antennas with additional isolation of near field arbitrary objects,” Electron. Lett., 1998; (ii)  “Reconfigurable antenna arrays with multiple requirements: A versatile 3D approach,” Int. J. Antennas Propag., 2017; (iii) “Power pattern synthesis of reconfigurable conformal arrays with near-field constraints,” IEEE Trans. Antennas Propag., 2004; (iv) “Synthesis of antenna patterns with imposed near-field nulls,” Electron. Lett., 2006; .... The authors should insert these (and other) studies and discuss their contribution with respect them.

3) The application of the FDA-based method to sole square and circular array seems an unnecessary limitation. The method seems suitable for any geometry, thus a generalization in this sense may improve the overall contribution of the work.

4) The synthesis accounts for the sole array factor and not for the element factor. This does not allows one to consider the impact of mutual coupling, an effect that current antenna designers cannot neglect. In the provided figures showing the arrays, the authors given the idea of having assumed microstrips, which are really sensitive to mutual coupling, Thus, the authors should provide at least an example where mutual coupling is considered.

5) Some figures are of not sufficient quality. While Fig. 12, for example, gives the idea of being directly saved from the software where it has been created (and is hence of satisfactory quality), many others (Figs. 2, 3, 5,6, 9, 10, 11) seems taken from another documents and hence have some shadows and not neat views.

Reviewer 2 Report

In this work, the authors developed a numerical optimization method to achieve the near-field focusing of square arrays and circular arrays in radio frequencies. It is shown that to change the initial amplitude and working frequency, the proposed approach operates based on the frequency diversity array technique. Their results revealed the results of two different kinds of arrays can be verified by numerical methods and full-wave simulation. Although the work contains some new results, it suffers from lack of important and fundamental calculations as well as presentation drawbacks. I listed my comments below and the authors may address the concerns in the revisions.

1) The referencing quality of the work is weak. The authors have to consider the recent state-of-the art publications in their introduction. This will improve the quality of the article and shows that the authors have conducted a deep literature search.

2) all acronyms must be demonstrated in their expanded form, such as "WPT".

3) The Figure captions are too brief. All figures require detailed description. 

4) What is the influence of the shape of resonators on the near-field focusing performance?

5) In Figure 8, the quality-factor and dephasing time of the induced resonant lineshape is missed. These two parameters can be quantified using the proposed approach by Nanoscale 11 (17), 8091-8095 (2019).

Round 2

Reviewer 1 Report

The paper is the revised version of a previously submitted manuscript in which the authors propose a numerical optimization method for the near-field synthesis of planar antenna arrays adopting  the frequency diversity array (FDA) approach. With respect to the previous version, the main issues raised by this referee in terms of content and presentation have been addressed.

Author Response

The main issues raised by this referee  have been addressed.

Reviewer 2 Report

Although the authors tried ti improve the quality of the article in the revisions, there is still a critical point that must be addressed. In my previous review, I requested for the quantification of dephasing time using the proposed method there. The authors should note that the utilized method in the revisions is not appropriate. Again, I do suggest them to consider the following works for this purpose:

Nanoscale 11 (17), 8091-8095 (2019), and Physical Review Letters, 80(19), 4249 (1998).

Round 3

Reviewer 2 Report

Publishable as is.